# Equivalence between representational similarity analysis, centered kernel alignment, and canonical correlations analysis

**Alex H. Williams**
Center for Neural Science, New York University
Center for Computational Neuroscience, Flatiron Institute
`alex.h.williams@nyu.edu`

## Abstract

Centered kernel alignment (CKA) and representational similarity analysis (RSA) of dissimilarity matrices are two popular methods for comparing neural systems in terms of representational geometry. Although they follow a conceptually similar approach, typical implementations of CKA and RSA tend to result in numerically different outcomes. Here, I show that these two approaches are largely equivalent once one incorporates a mean-centering step into RSA. This equivalence holds for both linear and nonlinear variants of these methods. These connections are simple to derive, but appear to have been thus far overlooked in the context of comparing neural representations. By unifying these measures, this paper hopes to simplify a complex and fragmented literature on this subject.

## 1 Introduction

Representational similarity analysis (RSA) is a decades-old framework for comparing neural response patterns across systems. It was developed in the cognitive science and computational neuroscience communities, within which it remains a very popular technique [25, 17, 24, 13]. While the RSA framework is quite general, most practical applications involve the construction of *representational dissimilarity matrices* (RDMs). For an experiment where neural responses are measured across $M$ conditions, an RDM is a symmetric $M \times M$ matrix that captures response dissimilarities across all unique condition pairs. Similarity between networks is quantified by similarity in their RDMs (e.g. by cosine similarity or Pearson correlation). I refer to this class of methods as RDM-RSA.

RDM-RSA bears a close resemblance to centered kernel alignment (CKA), a more recently developed framework for comparing neural representations [6, 23]. CKA is massively popular within the deep learning community, garnering over 1250 citations as of the time of this writing. In place of RDMs, CKA constructs $M \times M$ kernel matrices that capture neural response similarities through a positive definite kernel function. CKA quantifies similarity between networks as similarity in their kernel matrices, in exact analogy to how RSA quantifies similarity between RDMs.

To what extent are these methods quantifying the same thing? Here, I document several connections:

- First, a popular variant of RDM-RSA is to use squared Euclidean distance to construct RDMs and then use cosine similarity to compare RDMs [45]. I show that if one applies a centering operation before comparing the RDMs, the result is identical to linear CKA, which is the most popular variant of CKA.

- It is also popular to construct RDMs with Mahalanobis distance [45]. Here, I show that incorporating the centering operation on RDMs leads to a connection with canonical correlations analysis (CCA). Specifically, under a particular choice of covariance matrix, the

centered Mahalanobis RSA score equals the mean squared canonical correlation—a quantity sometimes called *Yanai's generalized coefficient of determination* [31, 47].

- Finally, I comment on nonlinear extensions of CKA and RSA. In CKA, this is achieved by using nonlinear kernel functions [23], while a nonlinear extension of RDM-RSA was recently introduced by Lin and Kriegeskorte [26]. I point out a simple way to construct an RDM using a nonlinear kernel function. Here again, RSA on the centered RDM yields the same result as CKA. This new approach conceptually mirrors that of Lin and Kriegeskorte [26]. Thus, nonlinear variants of CKA and RSA are also quite similar. Moreover, I note that performing RSA on centered Euclidean RDMs (instead of squared Euclidean RDMs) is equivalent to a form of nonlinear CKA.

These relationships are straightforward to derive, but I have not seen them laid out explicitly in the context of comparing neural representations. Kornblith et al. [23] documented a relationship between linear CKA and CCA, which will be leveraged as part of point 2 above. Additionally, Diedrichsen et al. [8] describe a relationship between linear CKA and RSA with *whitened* RDMs in the presence of independent and identically distributed measurement noise. The relationships between RSA and CKA I describe here are more basic—in fact, my exposition will treat neural responses as noise-free. Finally, Sejdinovic et al. [37] characterize the equivalence of kernel-based and distance-based hypothesis tests for variable independence. Similar results also appear in the context of multidimensional scaling algorithms [5] and broader mathematical literature [34]. However, it is easy for practitioners to overlook this prior literature because *(a)* existing papers are focused on distinct motivating applications, and because *(b)* many presentations assume an audience with strong mathematical background. I therefore believe there is utility in digesting and interpreting these results in plain terms to the neuroscience and interpretable AI communities.

## 2 Background

### 2.1 Summary of RDM-RSA

RSA is motivated by decades-old concepts from psychology and philosophy. In particular, Shepard and Chipman [38] posited that similar external objects (e.g. a square and rectangle) are mapped onto mental representations that are also, in some sense, close together—more so than mental representations of dissimilar external objects (e.g. a square and a cat). Or, as Edelman [12] succinctly put it, *"Representation is representation of similarities."*

RDM-RSA is a quantitative framework that enables practitioners to concretely apply these concepts to neural data analysis [24]. Formally, given $M$ stimulus conditions and $N$-dimensional neural response vectors $\boldsymbol{x}_1, \ldots, \boldsymbol{x}_M \in \mathbb{R}^{N_X}$, the first step of RDM-RSA is to compute an $M \times M$ *representational dissimilarity matrix (RDM)*. For example, if the squared Euclidean distance is used to compare neural responses, the elements of the RDM will be given by: $\boldsymbol{D}_{ij}^X = \|\boldsymbol{x}_i - \boldsymbol{x}_j\|_2^2$. Then, given responses from a second system, a second RDM, $\boldsymbol{D}_{ij}^Y = \|\boldsymbol{y}_i - \boldsymbol{y}_j\|_2^2$, is computed from neural responses $\boldsymbol{y}_1, \ldots, \boldsymbol{y}_M \in \mathbb{R}^{N_Y}$ across the same $M$ stimulus conditions. Finally, similarity between the RDMs is quantified by, for example, computing the Spearman or Pearson correlation between the upper triangular elements of $\boldsymbol{D}^X$ and $\boldsymbol{D}^Y$. The overall workflow can be summarized as follows:

---

**Procedure to compute RDM-RSA similarity scores:**
- Given neural responses, $\boldsymbol{x}_1, \ldots, \boldsymbol{x}_M \in \mathbb{R}^{N_X}$ and $\boldsymbol{y}_1, \ldots, \boldsymbol{y}_M \in \mathbb{R}^{N_Y}$.
- Specify distance functions, $d^X$ and $d^Y$.
- Specify an RDM comparison function, $s : \mathbb{R}^{M \times M} \times \mathbb{R}^{M \times M} \mapsto \mathbb{R}_+$.
- Compute RDMs, $\boldsymbol{D}_{ij}^X = d^X(\boldsymbol{x}_i, \boldsymbol{x}_j)$ and $\boldsymbol{D}_{ij}^Y = d^Y(\boldsymbol{y}_i, \boldsymbol{y}_j)$.
- Finally, compute the RDM similarity $S(\boldsymbol{D}^X, \boldsymbol{D}^Y)$.

---

Throughout this paper we will use cosine similarity to compare RDMs. Thus we define:

$$S(\boldsymbol{A}, \boldsymbol{B}) = \frac{\mathrm{Tr}[\boldsymbol{A}\boldsymbol{B}]}{\|\boldsymbol{A}\|_F \|\boldsymbol{B}\|_F} = \frac{\mathrm{vec}(\boldsymbol{A})^\top \mathrm{vec}(\boldsymbol{B})}{\|\mathrm{vec}(\boldsymbol{A})\|_2 \|\mathrm{vec}(\boldsymbol{B})\|_2} \tag{1}$$

as the comparison function between any two symmetric matrices $\boldsymbol{A} \in \mathbb{R}^{M \times M}$ and $\boldsymbol{B} \in \mathbb{R}^{M \times M}$.

## 2.2 Summary of CKA

In machine learning, there has been a swell of recent interest on the topic of comparing neural representations across networks. Popular approaches include centered kernel alignment (CKA; [6, 23]), canonical correlations analysis (CCA; [30]), and Procrustes shape distance [9, 46].

Of these approaches, CKA is most obviously similar to RDM-RSA. In fact, one can consider CKA to be a special case of RSA that involves computing and comparing *representational similarity matrices* (RSMs) instead of RDMs. However, both historically and in current practice, RSMs are used considerably less frequently than RDMs by cognitive neuroscientists and psychologists. Thus, I have focused the narrative of this paper on RDM-RSA to construct a foil of CKA.

Before describing the procedure for computing CKA, two definitions must be introduced. First, a **positive definite kernel** function is, informally, a similarity function that, when applied pairwise to a set of $M$ neural response patterns, is guaranteed to produce a symmetric $M \times M$ matrix with nonnegative eigenvalues (i.e. a positive semidefinite matrix). Positive definite kernels are fundamental to modern machine learning theory, and a more deep and formal treatment is provided, for example, by [39]. We will use $k^X$ and $k^Y$ to denote positive definite kernels, and we will see that these functions play an analogous role to the distance/dissimilarity functions $d^X$ and $d^Y$ in RDM-RSA.

Next, the $M \times M$ **centering matrix** as is a matrix given by $\boldsymbol{C} = \boldsymbol{I} - \frac{1}{M}\mathbf{1}\mathbf{1}^\top$ where $\mathbf{1} \in \mathbb{R}^M$ is a vector full of ones. The reader can verify that multiplying any $M \times N$ matrix on the left by $\boldsymbol{C}$ results produces another $M \times N$ matrix whose columns sum to zero. Furthermore, for a symmetric matrix $\boldsymbol{A} \in \mathbb{R}^{M \times M}$ the centered matrix $\boldsymbol{C}\boldsymbol{A}\boldsymbol{C}$ has rows and columns that sum to zero. Moreover, $\sum_{ij}[\boldsymbol{C}\boldsymbol{A}\boldsymbol{C}]_{ij} = 0$.

With these definitions in hand, we are ready to state the procedure for computing CKA scores.

> **Procedure to compute CKA similarity scores:**
> - Given neural responses, $\boldsymbol{x}_1, \ldots, \boldsymbol{x}_M \in \mathbb{R}^{N_X}$ and $\boldsymbol{y}_1, \ldots, \boldsymbol{y}_M \in \mathbb{R}^{N_Y}$.
> - Specify positive definite kernel functions, $k^X$ and $k^Y$.
> - Compute kernel matrices, $\boldsymbol{K}^X_{ij} = k^X(\boldsymbol{x}_i, \boldsymbol{x}_j)$ and $\boldsymbol{K}^Y_{ij} = k^Y(\boldsymbol{y}_i, \boldsymbol{y}_j)$.
> - Compute *centered* cosine similarity, $S(\boldsymbol{C}\boldsymbol{K}^X\boldsymbol{C}, \boldsymbol{C}\boldsymbol{K}^Y\boldsymbol{C})$.

The kernel matrices $\boldsymbol{K}^X$ and $\boldsymbol{K}^X$ in CKA are analogous to the RDMs $\boldsymbol{D}^X$ and $\boldsymbol{D}^Y$. Furthermore, the kernel matrices are guaranteed to be positive semidefinite (i.e. be symmetric with nonnegative eigenvalues)—this guarantee comes from our definition of positive definite kernel functions, given above. The *centered* kernel matrices $\boldsymbol{C}\boldsymbol{K}^X\boldsymbol{C}$ and $\boldsymbol{C}\boldsymbol{K}^Y\boldsymbol{C}$ are also positive semidefinite because: *(a)* the centering matrix, $\boldsymbol{C}$, is positive semidefinite, and *(b)* positive semidefinite matrices are closed under matrix multiplication.

It may not be entirely obvious to some readers why the entries of $\boldsymbol{K}^X$ and $\boldsymbol{K}^Y$ should be interpreted as *similarity scores* between neural population responses. This is due to a result called Mercer's theorem which states, in essence, that any positive definite kernel can be interpreted as an inner product on some feature space (see [39] for a more rigorous introduction). Inner products are a measure of similarity—they increase as the angle between vectors decreases (i.e. as the vectors become more aligned with each other).

To summarize, there are two major differences between RDM-RSA and CKA. First, in place of RDMs, CKA uses *kernel matrices* $\boldsymbol{K}^X_{ij}$ and $\boldsymbol{K}^Y_{ij}$ which can be interpreted as $M \times M$ representational similarity (instead of dissimilarity) matrices. Second, to quantify similarity between kernel matrices, CKA computes the cosine similarity between *centered* kernel matrices. This centering step is typically absent in implementations of RSA, which will turn out to be critical.

## 2.3 Summary of CCA

Canonical correlation analysis (CCA) is a classical multivariate analysis method that identifies a sequence of maximally correlated one-dimensional projections from a pair of datasets [18]. When applying CCA to neural representations, $\boldsymbol{x}_1, \ldots, \boldsymbol{x}_M \in \mathbb{R}^{N_X}$ and $\boldsymbol{y}_1, \ldots, \boldsymbol{y}_M \in \mathbb{R}^{N_Y}$ as introduced above, the outcome of CCA will be a sequence of $N$ *canonical correlation coefficients* $1 \geq \rho_1 \geq \rho_2 \geq \cdots \geq \rho_N \geq 0$ where $N = \min(N_X, N_Y)$. The canonical correlations are defined as the solutions to sequence of optimization problems. The top coefficient, $\rho_1$, is given by:

$$
\begin{aligned}
\underset{\boldsymbol{w}, \boldsymbol{h}}{\text{maximize}} \quad & \sum_i \boldsymbol{w}^\top (\boldsymbol{x}_i - \bar{\boldsymbol{x}}) \cdot \boldsymbol{h}^\top (\boldsymbol{y}_i - \bar{\boldsymbol{y}}) \\
\text{subject to} \quad & \sum_i (\boldsymbol{w}^\top (\boldsymbol{x}_i - \bar{\boldsymbol{x}}))^2 = \sum_i (\boldsymbol{h}^\top (\boldsymbol{y}_i - \bar{\boldsymbol{y}}))^2 = 1
\end{aligned}
\tag{2}
$$

where $\boldsymbol{w} \in \mathbb{R}^{N_X}$, $\boldsymbol{h} \in \mathbb{R}^{N_Y}$ parameterize linear projections, $\bar{\boldsymbol{x}} = \frac{1}{M} \sum_i \boldsymbol{x}_i$ denotes the mean neural response for the first system, and $\bar{\boldsymbol{y}} = \frac{1}{M} \sum_i \boldsymbol{y}_i$ denotes the mean neural response of the second system. Subsequent canonical correlation coefficients are found by solving the same optimization problem subject to an appropriate orthogonality constraint on the projection vectors. See Eaton [11] for more detailed background.

Larger canonical correlation coefficients indicate greater alignment between neural representations, and past work in both machine learning [30, 27] and neuroscience [41, 14] has used CCA as a framework for comparing representations across neural systems. The average canonical correlation, $\frac{1}{N} \sum_i \rho_i$, and the average squared canonical correlation, $\frac{1}{N} \sum_i \rho_i^2$, can be used to summarize the overall similarity between two multivariate datasets [47, 30, 46].

Superficially, CCA does not ressemble RSA or CKA. But it turns out that they can be related by a simple change of variables. Specifically, let $\boldsymbol{\Sigma}_X$ and $\boldsymbol{\Sigma}_Y$ denote the $N_X \times N_X$ and $N_Y \times N_Y$ covariance matrices across the $M$ stimulus conditions within each neural system:

$$
\boldsymbol{\Sigma}_X = \frac{1}{M} \sum_i (\boldsymbol{x}_i - \bar{\boldsymbol{x}})(\boldsymbol{x}_i - \bar{\boldsymbol{x}})^\top \quad \text{and} \quad \boldsymbol{\Sigma}_Y = \frac{1}{M} \sum_i (\boldsymbol{y}_i - \bar{\boldsymbol{y}})(\boldsymbol{y}_i - \bar{\boldsymbol{y}})^\top
\tag{3}
$$

Then we define a linearly transformed set of responses as:

$$
\widetilde{\boldsymbol{x}}_i = \boldsymbol{\Sigma}_X^{-1/2} \boldsymbol{x}_i \quad \text{and} \quad \widetilde{\boldsymbol{y}}_i = \boldsymbol{\Sigma}_Y^{-1/2} \boldsymbol{y}_i
\tag{4}
$$

for $i = 1, \ldots, M$. It is common to refer to this change of variables as a *whitening transformation* (see e.g. [1]). Note that the whitening transformation assumes that $\boldsymbol{\Sigma}_X$ and $\boldsymbol{\Sigma}_Y$ are invertible; it is possible to incorporate regularization into this change of variables and achieve similar results.

The following lemma states that performing linear CKA on the transformed variables yields $\frac{1}{N} \sum_i \rho_i^2$ as a similarity measure. This was previously noted by Kornblith et al. [23]; the lemma below is only a slight reformulation of the statement in their paper.

**Lemma 1.** *When using linear kernel matrices on whitened neural responses, $\boldsymbol{K}_{ij}^X = \widetilde{\boldsymbol{x}}_i^\top \widetilde{\boldsymbol{x}}_j$ and $\boldsymbol{K}_{ij}^Y = \widetilde{\boldsymbol{y}}_i^\top \widetilde{\boldsymbol{y}}_j$, the CKA similarity score is equal to the average squared canonical correlation coefficient between $\{\boldsymbol{x}_1, \ldots, \boldsymbol{x}_M\}$ and $\{\boldsymbol{y}_1, \ldots, \boldsymbol{y}_M\}$. That is,*

$$
S(\boldsymbol{C}\boldsymbol{K}^X\boldsymbol{C}, \boldsymbol{C}\boldsymbol{K}^Y\boldsymbol{C}) = \frac{1}{N} \sum_i \rho_i^2
\tag{5}
$$

We will make use of this relationship in section 3.2 to show that performing RSA with centered squared Mahalanobis RDMs also yields $\frac{1}{N} \sum_i \rho_i^2$ as a measure of network similarity.

## 3  Results

It is clear that RDM-RSA and CKA are conceptually similar methods, but do they yield quantitatively similar outcomes? I now document several instances where they coincide *exactly*. All of these are special instances of the following result, stated formally below as proposition 1.

**Proposition 1.** *Let $k^X$ and $k^Y$ be positive definite kernel functions associated with kernel matrices:*

$$\boldsymbol{K}_{ij}^X = k^X(\boldsymbol{x}_i, \boldsymbol{x}_j) \qquad and \qquad \boldsymbol{K}_{ij}^Y = k^Y(\boldsymbol{y}_i, \boldsymbol{y}_j) \tag{6}$$

*Further, let $\boldsymbol{D}^X$ and $\boldsymbol{D}^Y$ be RDMs defined in terms of this kernel function:*

$$\boldsymbol{D}_{ij}^X = \boldsymbol{K}_{ii}^X + \boldsymbol{K}_{jj}^X - 2\boldsymbol{K}_{ij}^X \qquad and \qquad \boldsymbol{D}_{ij}^Y = \boldsymbol{K}_{ii}^Y + \boldsymbol{K}_{jj}^Y - 2\boldsymbol{K}_{ij}^Y \tag{7}$$

*Then, the centered cosine similarity scores between these matrices agree:*

$$S(\boldsymbol{C}\boldsymbol{D}^X\boldsymbol{C}, \boldsymbol{C}\boldsymbol{D}^Y\boldsymbol{C}) = S(\boldsymbol{C}\boldsymbol{K}^X\boldsymbol{C}, \boldsymbol{C}\boldsymbol{K}^Y\boldsymbol{C}) \tag{8}$$

This result follows from straightforward algebraic manipulations. A step-by-step derivation is provided in appendix A. As mentioned in section 1, equivalences between distance metrics and kernels are already established in the broader literature [34, 4, 37], but they appear to be overlooked, or at least underappreciated, within the context of comparing neural representational geometry.

The rest of this section discusses three specific cases of interest. In each, proposition 1 is used to show a near equivalence between a popular form of RDM-RSA and CKA or CCA.

### 3.1 Equivalence of Linear CKA and Squared Euclidean RDM-RSA

We first consider RDMs that are constructed using the squared Euclidean distance:

$$\boldsymbol{D}_{ij}^X = \|\boldsymbol{x}_i - \boldsymbol{x}_j\|_2^2 \qquad and \qquad \boldsymbol{D}_{ij}^Y = \|\boldsymbol{y}_i - \boldsymbol{y}_j\|_2^2 \tag{9}$$

Since $\|\boldsymbol{x}_i - \boldsymbol{x}_j\|_2^2 = \boldsymbol{x}_i^\top \boldsymbol{x}_i + \boldsymbol{x}_j^\top \boldsymbol{x}_j - 2\boldsymbol{x}_i^\top \boldsymbol{x}_j$, we see that eq. (7) implies that the corresponding kernel matrices are given by:

$$\boldsymbol{K}_{ij}^X = \boldsymbol{x}_i^\top \boldsymbol{x}_j \qquad and \qquad \boldsymbol{K}_{ij}^Y = \boldsymbol{y}_i^\top \boldsymbol{y}_j \tag{10}$$

The kernel function associated with these matrices is called the *linear kernel*, and the CKA score between linear kernel matrices is called *linear CKA*. Proposition 1 implies that performing RSA on the centered RDMs in eq. (9) is an equivalent to performing CKA on the kernel matrices in eq. (10).

We remark that this choice of distance (squared Euclidean) and kernel function (linear) are among the most popular variants of RDM-RSA and CKA, respectively. As of this writing, the squared Euclidean distance is currently the default option for constructing an RDM in the `rsatoolbox` Python package [33]. Furthermore, recent work has leveraged the mathematical tractability of squared Euclidean RDMs to establish statistical inference frameworks for RSA [36, 8]. Similarly, the predominant form of CKA within the deep learning community uses linear kernels eq. (10). Indeed, the paper popularizing CKA advocated explicitly for using linear kernels as a default choice [23]. Given the popularity of these two methods, it is somewhat surprising that their near equivalence has not been previously documented.

### 3.2 Equivalence of CCA and Mahalanobis RDM-RSA

Next, we consider RDMs that are constructed by the squared Mahalanobis distance. Formally, let $\boldsymbol{P}_X \in \mathbb{R}^{N \times N}$ and $\boldsymbol{P}_Y \in \mathbb{R}^{N \times N}$ be two arbitrary positive definite matrices and define:

$$\boldsymbol{D}_{ij}^X = (\boldsymbol{x}_i - \boldsymbol{x}_j)^\top \boldsymbol{P}_X^{-1}(\boldsymbol{x}_i - \boldsymbol{x}_j) \qquad and \qquad \boldsymbol{D}_{ij}^Y = (\boldsymbol{y}_i - \boldsymbol{y}_j)^\top \boldsymbol{P}_Y^{-1}(\boldsymbol{y}_i - \boldsymbol{y}_j). \tag{11}$$

as two RDMs. To achieve the desired relation in eq. (7), we choose the kernel matrices to be:

$$\boldsymbol{K}_{ij}^X = \boldsymbol{x}_i^\top \boldsymbol{P}_X^{-1} \boldsymbol{x}_j \qquad and \qquad \boldsymbol{K}_{ij}^Y = \boldsymbol{y}_i^\top \boldsymbol{P}_Y^{-1} \boldsymbol{y}_j \tag{12}$$

Proposition 1 implies that performing RSA on the centered RDMs in eq. (11) is an equivalent to performing CKA on the kernel matrices in eq. (12) for any choice of $\boldsymbol{P}_X$ and $\boldsymbol{P}_Y$.

Moreover, notice that the kernel matrices in eq. (12) can be interpreted as linear kernel matrices under the change of variables $\widetilde{\boldsymbol{x}}_i = \boldsymbol{P}_X^{-1/2} \boldsymbol{x}_i$ and $\widetilde{\boldsymbol{y}}_i = \boldsymbol{P}_Y^{-1/2} \boldsymbol{y}_i$. This change of variables corresponds to a whitening transformation when we choose, $\boldsymbol{P}_X = \boldsymbol{\Sigma}_X$ and $\boldsymbol{P}_Y = \boldsymbol{\Sigma}_Y$ using definitions from eq. (3). Thus, by lemma 1, the CKA score computed from the kernel matrices in eq. (12) is equal to the average squared canonical correlation coefficient. Therefore, by proposition 1, the cosine similarity RSA score computed from the centered RDMs in eq. (11) is also equal to this quantity.

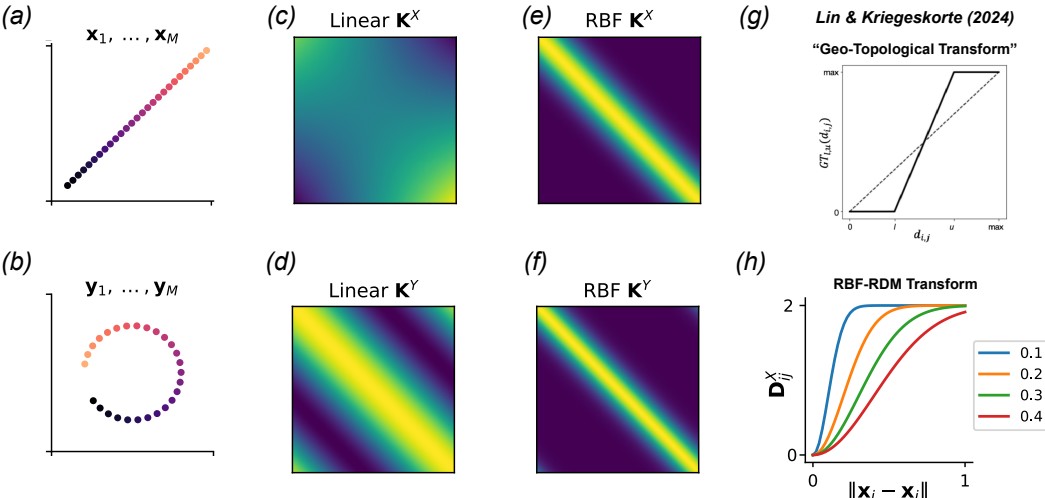

*Figure 1:* Nonlinear extensions of CKA and RSA. *(a-b)* Toy example of neural responses in $N_X = N_Y = 2$ dimensional space. Different colors correspond to matched stimulus conditions across the two point clouds. *(c-d)* Linear kernel matrices computed from the representations in panels (a) and (b). The network similarity is 0.595 according to linear CKA. *(e-f)* Nonlinear kernel matrices using RBF kernel functions with a bandwidth parameters of $\ell = 0.3$. The kernel matrices look more similar and indeed the network similarity score is higher, 0.982, according to this nonlinear extension of CKA. *(g)* Lin and Kriegeskorte [26] proposed a nonlinear extension of RSA in which RDMs are transformed elementwise by a monotonically increasing piecewise linear function. *(h)* When we translate the nonlinear CKA procedure in panels (e-f) into an equivalent RDM-RSA procedure according to proposition 1, we observe that the nonlinear RBF kernel induces a similar "geo-topological transform" on the Euclidean distances between neural responses. The shape of this transform is modulated by the bandwidth parameter, $\ell$, plotted as different colors.

The squared Mahalanobis distance is a popular method for constructing RDMs. It is supported by the `rsatoolbox` package [33] and discussed in multiple recent papers [45, 8, 36]. We must mention, however, a key difference between these works and our above analysis with respect to CCA. Typically, Mahalanobis RDMs are motivated by choosing $\boldsymbol{P}_X$ to be the covariance of "noise" in the neural response. To follow this motivation, $\boldsymbol{P}_X$ and $\boldsymbol{P}_Y$ are often set to the covariance matrices computed from residuals of a simple model [45]. The choice of $\boldsymbol{P}_X = \boldsymbol{\Sigma}_X$ and $\boldsymbol{P}_Y = \boldsymbol{\Sigma}_Y$ was made to elucidate a connection to CCA. This does not capture "noise" per se, as it applies equally well to a complete noiseless, deterministic system (e.g. a feedforward deep network). The choice of $\boldsymbol{P}_X = \boldsymbol{\Sigma}_X$ and $\boldsymbol{P}_Y = \boldsymbol{\Sigma}_Y$ is nonetheless similar in the sense that one could replace the average neural responses, $\bar{\boldsymbol{x}}$ and $\bar{\boldsymbol{y}}$ in eq. (3), with a condition-specific model prediction.

### 3.3 Equivalence of Nonlinear CKA and Topological RSA [26]

One of the nice features of CKA is that it nicely extends to nonlinear kernel functions. For example, the standard radial basis function (RBF) kernel (also known as the squared exponential kernel) is a positive definite kernel with a tuneable bandwidth parameter, $\ell$. Using this function leads to the nonlinear kernel matrices:

$$\boldsymbol{K}_{ij}^X = \exp\left(-\frac{\|\boldsymbol{x}_i - \boldsymbol{x}_j\|_2^2}{2\ell_X^2}\right) \qquad \text{and} \qquad \boldsymbol{K}_{ij}^Y = \exp\left(-\frac{\|\boldsymbol{y}_i - \boldsymbol{y}_j\|_2^2}{2\ell_Y^2}\right) \qquad (13)$$

where we have allowed for the possibility of separating tuning different bandwidth parameters, $\ell_X$ and $\ell_Y$, for each network. Kornblith et al. [23] briefly commented on using this approach to compute nonlinear CKA similarity scores, but only a few works have seriously followed up on this possibility [2].

A practitioner may be interested in using nonlinear CKA to achieve a similarity measure between neural representations that have dissimilar *shapes* but have similar topological features. More precisely, nonlinear CKA using RBF kernel matrices will characterize neural representations as similar when short-range distances between neural responses are preserved, but long-range distances

are potentially quite different. This results in a similarity measure that is mostly insensitive to continuous deformations that do not change the topology such as bending. A toy example where nonlinear CKA succeeds at capturing topological similarities, but linear CKA fails to do so, is shown in Figure 1a-f. The idea of developing metrics that capture topological (and not geometric) similarity between neural representations has garnered recent interest within the community, but requires more research to be fully fleshed out and understood [26, 29, 3, 19].

Interestingly, Lin and Kriegeskorte [26] recently proposed a nonlinear extension to RDM-RSA that involves applying an monotonic and saturating transform to the elements of an RDM (see Figure 1g). They show that this results in similarity measures that are sensitive to topological features of the neural representation. It is easy to see that this procedure is closely related to nonlinear CKA. In particular, the form of the squared exponential kernel means that $\boldsymbol{K}_{ii}^X = 1$ and $\boldsymbol{K}_{ii}^Y = 1$ for all $i = 1, \ldots, M$. Thus, by eq. (7), the nonlinear RDMs associated with the RBF kernel take the form:

$$\boldsymbol{D}_{ij}^X = 2 - 2\boldsymbol{K}_{ij}^X \qquad \text{and} \qquad \boldsymbol{D}_{ij}^Y = 2 - 2\boldsymbol{K}_{ij}^Y \qquad (14)$$

Inspecting these expressions carefully, we realize that $\boldsymbol{D}_{ij}^X$ and $\boldsymbol{D}_{ij}^Y$ are saturating, monotonically increasing functions of the squared Euclidean distances, $\|\boldsymbol{x}_i - \boldsymbol{x}_j\|_2^2$ and $\|\boldsymbol{y}_i - \boldsymbol{y}_j\|_2^2$ respectively. The bandwidth parameter $\ell^2$ controls the steepness of these nonlinear functions, as shown in Figure 1h.

In summary, a nonlinear kernel can be used to construct a RDM using eq. (7). This construction of an RDM via a nonlinear kernel can be viewed as applying an elementwise nonlinearity to a squared Euclidean RDM (as in eq. 14 for the case of a RBF kernel). By proposition 1, the centered cosine similarity between nonlinearly transformed RDMs will be equivalent to performing nonlinear CKA. Qualitatively, this approach resembles the topological RSA method introduced by Lin and Kriegeskorte [26].

### 3.4 RSA on Euclidean RDMs is also a form of nonlinear CKA

Equation (7) shows how we can use a positive definite kernel function to create a notion of distance for which RDM-RSA (with centering and cosine similarity comparison) is equivalent to CKA. It is also possible to go on in the other direction—i.e. we can use a distance function[1] to define a positive definite kernel. For instance, consider the possibility of constructing RDMs using a fractional Euclidean distance:

$$\boldsymbol{D}_{ij}^X = \|\boldsymbol{x}_i - \boldsymbol{x}_j\|_2^q \qquad \text{and} \qquad \boldsymbol{D}_{ij}^Y = \|\boldsymbol{y}_i - \boldsymbol{y}_j\|_2^q \qquad (15)$$

where $0 < q \le 2$ is a user-defined hyperparameter. Of course, when $q = 2$ we recover the squared Euclidean distance, for which RDM-RSA is equivalent to linear CKA. However, the choice of $q = 1$ (i.e. the classic Euclidean distance) is also popular within the RSA literature and it may not be immediately clear how to map this onto a form of CKA.

It turns out that computing the centered RDM-RSA score with the distance matrices in eq. (15) is equivalent to performing CKA on the following kernel matrices (see, e.g., Example 15 in [37]):

$$\boldsymbol{K}_{ij}^X = \tfrac{1}{2}\left(\|\boldsymbol{x}_i\|_2^q + \|\boldsymbol{x}_i\|_2^q - \|\boldsymbol{x}_i - \boldsymbol{x}_j\|_2^q\right) \quad \text{and} \quad \boldsymbol{K}_{ij}^Y = \tfrac{1}{2}\left(\|\boldsymbol{y}_i\|_2^q + \|\boldsymbol{y}_i\|_2^q - \|\boldsymbol{y}_i - \boldsymbol{y}_j\|_2^q\right) \quad (16)$$

Indeed, it is easy to verify that eqs. (15) and (16) satisfy the relationship in eq. (7), which is the main requirement of proposition 1. It is less obvious that the expressions in eq. (16) define positive definite kernels, but it can be shown that they correspond to the covariance function of fractional Brownian motion [28], which is positive definite. This reduces to classical Brownian motion (or a Wiener process) when $q = 1$, which is a popular choice within RSA literature.

Interestingly, tuning the parameter $0 < q \le 2$ can result in a family of nonlinear similarity scores similar to the nonlinear RBF kernel with different bandwidth parameters, $\ell$, that was discussed above in section 3.3. To show this, Figure 2 revisits the toy example shown in Figure 1a-b, using fractional Euclidean distance RDMs in place of RBF kernel matrices. Figure 2a visualizes the raw RDMs, $\boldsymbol{D}^X$ and $\boldsymbol{D}^Y$, for various values of $q$. Qualitatively, we see that these RDMs appear more similar as $q$ decreases. A similar trend is seen in the centered RDMs, shown in Figure 2b.

---

[1]More precisely, the distance function must be a semimetric of negative type. See lemma 12 in Sejdinovic et al. [37] for a formal statement.

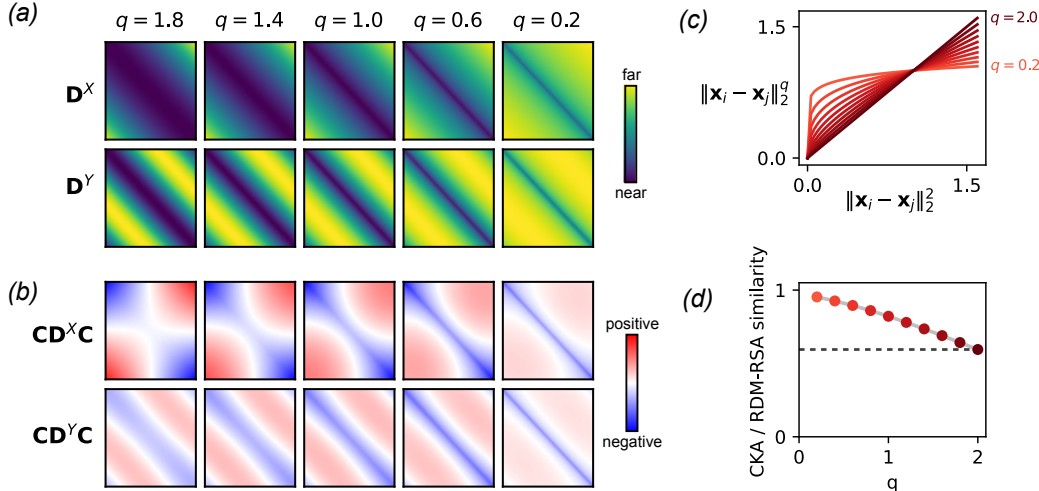

*Figure 2:* RSA on Euclidean RDMs is a form of nonlinear CKA. *(a)* RDMs computed from the toy example in Figure 1a-b using fractional Euclidean distance, eq. (15), for various choices of $0 < q \leq 2$. *(b)* Centered RDM matrices from panel *a*. By proposition 1, these centered RDMs are equal to negative two times the centered kernel matrices given in eq. (16). *(c)* The hyperparameter $q$ can be interpreted as applying an elementwise, monotonically increasing function to the squared Euclidean RDM (similar to the distance induced by the RBF kernel in Figure 1h). *(d)* The cosine similarity between $CD^X C$ and $CD^Y C$, which is equivalent to the CKA score on the kernel matrices in eq. (16), is plotted as a function of $q$. When $q = 2$, this converges to the linear CKA score (shown as black dashed line). Smaller values of $q$ result in higher similarity scores, emphasizing topological similarity between the response patterns in Figure 1a-b.

Intuitively, the fractional exponent $q$ applies an elementwise nonlinearity to the squared Euclidean RDM matrices (Figure 2c), which is similar to the RBF-RDM transform previously highlighted in Figure 2h. In the limit that $q \to 0$, the nonlinearity is a step function, equal to one everywhere except zero. Because of this step function behavior, the cosine similarity between centered RDMs will equal one in the limit that $q \to 0$ because every RDM will have zeros along the diagonal and ones on the off diagonals. On the other extreme, when $q = 2$, we recover squared Euclidean RDMs, and the resulting RDM-RSA score after centering will be equal to linear CKA. Intermediate values of $q$ will smoothly interpolate between these outcomes, as shown in Figure 2d.

In summary, this section has shown a new interpretation of RSA with Euclidean RDMs ($q = 1$) as a form of nonlinear CKA with a kernel function defined by Brownian motion or Wiener process. More generally, one can choose any value of $0 < q < 2$, which can be intereperted as a form of nonlinear CKA with a kernel related to fractional Brownian motion. When $q = 2$, we recover linear CKA or squared Euclidean RDM-RSA with centering.

## 4    Importance and Interpretation of Centering

We have seen that the key difference between CKA and commonly used RDM-RSA methods is the presence of the centering operation, $A \mapsto CAC$ for a symmetric marix $A$. Beyond deriving an equivalence between CKA and RDM-RSA, are there desirable reasons for this centering operation?

There is a simple justification for centering in the case of linear CKA. Specifically, consider translating the neural responses, $\widetilde{x}_i = x_i + \alpha$ and $\widetilde{y}_i = y_i + \beta$ for some arbitrary vectors $\alpha \in \mathbb{R}^{N_X}$ and $\beta \in \mathbb{R}^{N_Y}$. Translating the responses in this manner is akin to choosing a different origin for the coordinate system defining neural responses. It is easy to show that the linear CKA score computed on $\widetilde{x}_1, \ldots \widetilde{x}_M$ and $\widetilde{y}_1, \ldots \widetilde{y}_M$ is invariant to the value of $\alpha$ and $\beta$, while the *uncentered* cosine similarity can be made arbitrarily close to one. Thus, the centering operation on kernel matrices is necessary if one desires a translation-invariant measure of representational similarity. Readers seeking further intuition should take a closer look at Cortes et al. [6], who originally introduced the CKA score, and who argue that the centering step is "critical" at length in their paper.

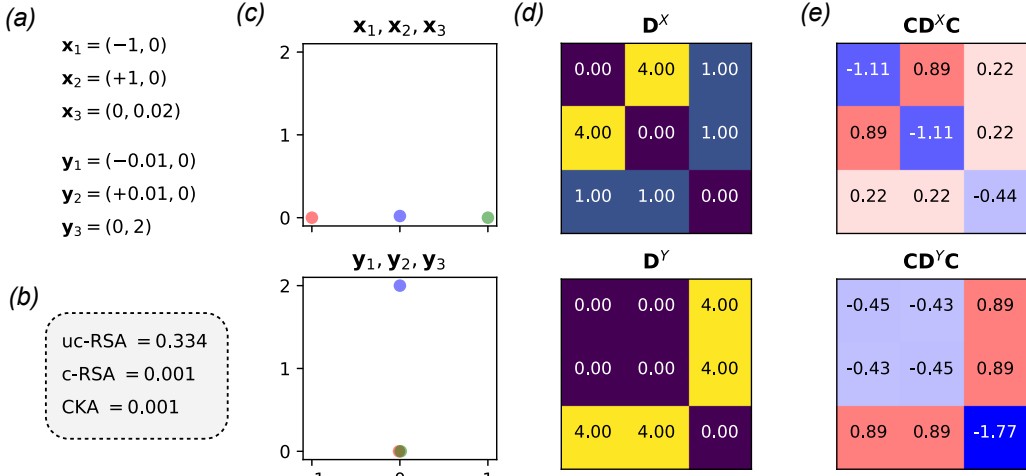

*Figure 3:* Intuition for centering operation on RDMs. **(a)** Explicit example of $M = 3$ neural response vectors in $N = 2$ dimensions. This toy example is illustrated for the rest of this figure. **(b)** Similarity scores for uncentered squared Euclidean RSA (uc-RSA), squared Euclidean RSA with centering (c-RSA), and linear CKA on the example activations given in panel $a$. Note that c-RSA and CKA give the same numeric value, as expected. Further, CKA and c-RSA are essentially zero, meaning that the two neural representations are "maximally dissimilar." **(c)** The top panel shows the 2D response vectors for the first system, $\{x_1, x_2, x_3\}$. The bottom panel shows the 2D response vectors for the second system, $\{y_1, y_2, y_3\}$. Red, green, and blue dots respectively denote the response on the first condition ($x_1$ and $y_1$), response on the second condition ($x_2$ and $y_2$), and response on the third condition ($x_3$ and $y_3$). **(d)** The $3 \times 3$ squared Euclidean RDMs associated with the two point configurations in panel $c$. **(e)** The same RDMs in panel $d$ after the centering operation. Negative entries are colored in blue and positive entries are colored in red.

The intuition behind centering RDMs is different. Unlike linear kernel matrices, the elements of RDMs are already invariant to the translations. That is, $\|\widetilde{x}_i - \widetilde{x}_j\|_2^2 = \|x_i - x_j\|_2^2$ for any transformation of the form $\widetilde{x}_i = x_i + \alpha$. On the other hand, because distances are nonnegative, the cosine similarity between uncentered RDMs can be inflated above zero. Incorporating the centering operation results in matrices with positive and negative entries, which intuitively can be "more orthogonal" resulting in cosine similarity scores closer to zero. Figure 3 illustrates this intuition in a simple toy example. Briefly, two neural systems are defined in $N = 2$ dimensions across $M = 3$ stimulus conditions (fig. 3a). The centered RSA and linear CKA scores are essentially zero, indicating that the two neural systems are maximally dissimilar; however, the uncentered RDM-RSA score is roughly $1/3$ (fig. 3b). In fact, these two triangular point configurations (visualized in fig. 3c) are maximally different shapes, as shown for example in [22]. Thus, in this setting where $M = 3$ conditions and $N = 2$ neural dimensions, uncentered RDM-RSA can only output a similarity score on the interval $[1/3, 1]$. Once centering is incorporated, the resulting score, equivalent to CKA, is normalized to lie on $[0, 1]$.

Applying the centering operation to distance matrices appears to have deeper importance in other contexts. For example, the distance covariance statistic [43, 42], which can be used to test for independence among random variables, is computed from centered distance matrices. One can show that removing the centering step is undesirable—the resulting statistic can fail to detect dependencies among random variables, effectively resulting in false negatives when hypothesis testing for independence. A concrete example of this failure is described in [32].

Centering is also included in distance covariance analysis [7], a method that leverages the distance covariance framework for dimensionality reduction. Kernel principal components analysis (kernel PCA) is a similar method to achieve nonlinear dimensionality reduction, and this too typically includes a centering step (see [35], appendix B). Intuitively, PCA fits a hyperplane passing through the origin to approximate high-dimensional data. Centering around the origin is therefore a sensible and important preprocessing step for this analysis.

The centering operation is also applied to squared Euclidean distance matrices in the context of multidimensional scaling [15, 5]. Here, the problem is to reconstruct a set of points, $\boldsymbol{x}_1, \ldots, \boldsymbol{x}_M$, that are consistent with a given pairwise distance matrix, $\boldsymbol{D}^X$. There are of course many solutions to this problem—any valid configuration of points can be freely rotated, reflected, and translated. If the configuration generating the squared distance matrix spans the full space, then one can perform an eigendecomposition of $\boldsymbol{C}\boldsymbol{D}^X\boldsymbol{C}$ to obtain a solution. Specifically, one obtains $\boldsymbol{C}\boldsymbol{D}^X\boldsymbol{C} = \boldsymbol{G}\boldsymbol{G}^\top$ where $\boldsymbol{G}$ is a matrix with orthogonal columns (scaled eigenvectors). The rows of $\boldsymbol{G}$ are taken as $M$ points in an $N_X$-dimensional space, and one can show that they indeed recover the appropriate pairwise Euclidean distance scores. Of course, $\boldsymbol{G}\boldsymbol{G}^\top$ is a linear kernel matrix, which is the core insight highlighted in this paper. Gower [15] remarks that the origin of the configuration will be the centroid of the points (i.e. a centered kernel matrix).

## 5  Conclusion

The contribution of this paper is to document some precise equivalences between two popular frameworks for quantifying representational similarity between neural systems: CKA and RDM-RSA. This was done by exploiting one-to-one relationships between positive kernel functions and distance functions that are already well-established in mathematical literature, tracing back to work by Schoenberg [34] (for more background, see [4]). Closely related work has highlighted similar equivalences within in the context of statistical tests for independence [37]. Nonetheless, to my best knowledge, the relationship between CKA and RDM-RSA has not been explicitly spelled out in prior work and is not widely understood by researchers in this area.

Indeed, while conceptual similarities between CKA and RDM-RSA are often acknowledged, they are mostly treated as being distinct methods (e.g. in [21]). Moreover, CKA and RDM-RSA are preferred by different research communities in machine learning and cognitive neuroscience for historical reasons. By illustrating deeper mathematical connections, I hope to encourage more exchanges and cross-citations between these communities. For example, Cortes et al. [6] provide a detailed theoretical analysis of CKA including results on bounds on how many sampled stimuli, $M$, are needed to achieve good estimates. Our analysis shows that these results can be immediately applied to RSA with centered squared Euclidean RDMs. Likewise, our results may also enable statistical frameworks developed for RDM-RSA (e.g. [8, 36]) to be adapted and applied to CKA-based analysis.

More broadly, the literature on comparing neural representations is complex and federated. A recent review by Klabunde et al. [20] catalogues over thirty methods for quantifying similarity. It is difficult for practitioners to choose among this large menu of options, many of which give different numerical outputs [40]. Cases where methods are truly identical ought to be widely appreciated and highlighted. Harvey et al. [16] previously showed that the Procrustes shape distances (advocated by [9, 46]) are equivalent to the normalized Bures similarity score (advocated in [44]) up to a monotonic transformation. This paper adds to this list by documenting several additional examples where RSA on centered RDMs coincides with linear CKA, CCA, and nonlinear CKA.

The analyses detailed in this paper focus on deterministic (i.e. noise-free) and static (i.e. non-dynamical) neural representations. This is a limitation, and there is growing interest within the literature to characterize the stochastic [10] and dynamical [29] aspects of neural representations. Future work that connects these emerging methodologies to CKA and RSA-based analyses would be of great interest.

## Acknowledgements

I am grateful to Nikolaus Kriegeskorte (Columbia University) for his comments, feedback, and encouragement during the writing of this manuscript.

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

# A Proof of proposition 1

Let $\langle \boldsymbol{K}^X \rangle_i = \frac{1}{M} \sum_j \boldsymbol{K}_{ij}^X$ denote the average of row $i$ in $\boldsymbol{K}^X$. Likewise, let $\langle \boldsymbol{D}^X \rangle_i = \frac{1}{M} \sum_j \boldsymbol{D}_{ij}^X$ denote the average of row $i$ in $\boldsymbol{D}^X$. By symmetry, note that $\langle \boldsymbol{K}^X \rangle_i$ and $\langle \boldsymbol{D}^X \rangle_i$ are also equal to the average of column $i$ in $\boldsymbol{K}^X$ and $\boldsymbol{D}^X$, respectively. Additionally, let $\langle\!\langle \boldsymbol{K}^X \rangle\!\rangle = \frac{1}{M^2} \sum_{ij} \boldsymbol{K}_{ij}^X$ denote the average element of $\boldsymbol{K}^X$. Likewise, let $\langle\!\langle \boldsymbol{D}^X \rangle\!\rangle = \frac{1}{M^2} \sum_{ij} \boldsymbol{D}_{ij}^X$ denote the average element of $\boldsymbol{D}^X$. Finally, we write the elements of the $M \times M$ centering matrix as $\boldsymbol{C}_{ij} = \delta_{ij} - \frac{1}{M}$, where $\delta_{ij} = 1$ if $i = j$ and equal to zero otherwise (i.e. the Kronecker delta function).

Using this notation, we can write the elements of the centered RDM as:

$$[\boldsymbol{C}\boldsymbol{D}^X\boldsymbol{C}]_{ij} = \sum_{k\ell} \boldsymbol{C}_{ik} \boldsymbol{D}_{k\ell}^X \boldsymbol{C}_{\ell j} \tag{17}$$

$$= \sum_{k\ell} (\delta_{ik} - \tfrac{1}{M}) \boldsymbol{D}_{k\ell}^X (\delta_{\ell j} - \tfrac{1}{M}) \tag{18}$$

$$= \sum_{k\ell} \delta_{ik}\delta_{\ell j} \boldsymbol{D}_{k\ell}^X - \frac{1}{M} \sum_{k\ell} \delta_{ik} \boldsymbol{D}_{k\ell}^X - \frac{1}{M} \sum_{k\ell} \delta_{\ell j} \boldsymbol{D}_{k\ell}^X + \frac{1}{M^2} \sum_{k\ell} \boldsymbol{D}_{k\ell}^X \tag{19}$$

$$= \boldsymbol{D}_{ij}^X - \langle \boldsymbol{D}^X \rangle_i - \langle \boldsymbol{D}^X \rangle_j + \langle\!\langle \boldsymbol{D}^X \rangle\!\rangle \tag{20}$$

Using identical algebriac manipulations, we see that the centered kernel matrix is given by:

$$[\boldsymbol{C}\boldsymbol{K}^X\boldsymbol{C}]_{ij} = \boldsymbol{K}_{ij}^X - \langle \boldsymbol{K}^X \rangle_i - \langle \boldsymbol{K}^X \rangle_j + \langle\!\langle \boldsymbol{K}^X \rangle\!\rangle \tag{21}$$

Now substitute in the definition of the RDM in terms of the the kernel matrix to achieve the following set of relations:

$$\boldsymbol{D}_{ij}^X = \boldsymbol{K}_{ii}^X + \boldsymbol{K}_{jj}^X - 2\boldsymbol{K}_{ij}^X \tag{22}$$

$$\langle \boldsymbol{D}^X \rangle_i = \boldsymbol{K}_{ii}^X + \frac{1}{M} \mathrm{Tr}[\boldsymbol{K}^X] - 2\langle \boldsymbol{K}^X \rangle_i \tag{23}$$

$$\langle \boldsymbol{D}^X \rangle_j = \frac{1}{M} \mathrm{Tr}[\boldsymbol{K}^X] + \boldsymbol{K}_{jj}^X - 2\langle \boldsymbol{K}^X \rangle_j \tag{24}$$

$$\langle\!\langle \boldsymbol{D}^X \rangle\!\rangle = \frac{2}{M} \mathrm{Tr}[\boldsymbol{K}^X] - 2\langle\!\langle \boldsymbol{K}^X \rangle\!\rangle \tag{25}$$

Plugging these four relationships into eq. (20) and simplifying yields:

$$[\boldsymbol{C}\boldsymbol{D}^X\boldsymbol{C}]_{ij} = -2\boldsymbol{K}_{ij}^X + 2\langle \boldsymbol{K}^X \rangle_i + 2\langle \boldsymbol{K}^X \rangle_j - 2\langle\!\langle \boldsymbol{K}^X \rangle\!\rangle = -2[\boldsymbol{C}\boldsymbol{K}^X\boldsymbol{C}]_{ij} \tag{26}$$

Thus, the centered RDM is equal to negative two times the centered kernel matrix. The proposition then immediately follows by recognizing that the cosine similarity function, defined in eq. (1), is invariant to this rescaling. That is, for any $c \neq 0$ and any symmetric matrices $\boldsymbol{A}$ and $\boldsymbol{B}$ we have:

$$S(c\boldsymbol{A}, c\boldsymbol{B}) = \frac{\mathrm{Tr}[c^2\boldsymbol{A}\boldsymbol{B}]}{\|c\boldsymbol{A}\|_F \|c\boldsymbol{B}\|_F} = \frac{c^2}{|c| \cdot |c|} S(\boldsymbol{A}, \boldsymbol{B}) = S(\boldsymbol{A}, \boldsymbol{B}) \tag{27}$$

Thus, we have:

$$S(\boldsymbol{C}\boldsymbol{D}^X\boldsymbol{C}, \boldsymbol{C}\boldsymbol{D}^Y\boldsymbol{C}) = S(-2 \cdot \boldsymbol{C}\boldsymbol{K}^X\boldsymbol{C}, -2 \cdot \boldsymbol{C}\boldsymbol{K}^Y\boldsymbol{C}) = S(\boldsymbol{C}\boldsymbol{K}^X\boldsymbol{C}, \boldsymbol{C}\boldsymbol{K}^Y\boldsymbol{C}) \tag{28}$$

as claimed by the proposition.

