# OpenReview forum: "Equivalence between representational similarity analysis, centered kernel alignment, and canonical correlations analysis"
_NeurIPS.cc/2024/Workshop/UniReps — UniReps_

### Official Review · Reviewer_HzJv · 2024-09-30
**The author's claim is simple and intuitive but appears incremental.**

**Rating:** 5
**Confidence:** 2

**Review:**

This paper establishes the equivalence between representational similarity analysis (RSA), centered kernel alignment (CKA), and canonical correlation analysis (CCA),  as the manuscript title states. CKA and RSA are popular techniques that can be applied to quantify the similarity between neural networks. The basis for proving the equivalence is CCA, a classical method for analyzing multivariate data; CCA finds the hidden correlation between two sets of variables. In the manuscript's setting, the two sets of variables correspond to the neural responses from different networks.

The author's central claim is Proposition 1, the equivalence between CCA and RSA. When calculating distance matrices and gram matrices from two neural responses, the centered cosine similarity (CCS) between the distance matrices is identical to that between the gram matrices. The arguments supporting this claim are quite clear because Proposition 1 is a straightforward application of Lemma 1 presented by Kornblith et al., which claims that the CCS between gram matrices is identical to the average of canonical correlations.

Pros:
- The main claim and supporting arguments are clear.

Cons:
- The main claim is straightforward and intuitive, but it is not so surprising. The manuscript's progress seems incremental compared to the importance of Lemma 1.

- The manuscript seems redundant compared to the simplicity of the main claim, and it becomes more concise and focused if sections 3.4 and 4 are moved to the appendix.

- No practical application was presented. Practitioners will get
more interested if the author sufficiently explains that CKA and RSA are
useful for quantifying similarity in neural representational geometry.

Minor comments:
- Section 2.3: The description might be misleading. Readers unfamiliar with CCA cannot readily understand that the optimizer of Eq. (2) becomes a ``sequence'' of canonical correlations. The section would read better if the author explained canonical correlations clearly. They should be obtained from the eigendecomposition of Sigma_{XX}^{-1} Sigma_{XY} Sigma_{YY}^{-1} Sigma_{YX}.

- page 3, line 94: The transpose symbol in the centering matrix is missing.

- First-person singular (I) and plural (we) are mixed.

---

### Official Review · Reviewer_j93Q · 2024-10-06
**A clear demonstration of cases where computing RDM-RSA is equivalent to CKA**

**Rating:** 9
**Confidence:** 5

**Review:**

Comparing the similarity between representations is a core problem in many fields such as Artificial Intelligence (AI), Neuroscience & Cognitive Science. However, different fields have different preferences for the metrics they commonly use. For example, the field of AI uses Centered Kernel Alignment (CKA), Neuroscience uses Representational Similarity Analysis (RSA), and Cognitive Science uses Canonical Correlation Analysis (CCA). This paper demonstrates and provides mathematical proofs of the cases where RSA, CKA, and CCA are equivalent. A mean-centering step is crucial in RSA for this equivalence, providing benefits such as translational invariance and testing for independence. Demonstrating this is important to enable cross-talk between the fields and can accelerate the development of such measures through interdisciplinary research.

The authors show three cases where RSA is equivalent to computing CKA (and CCA in some cases). Note that cosine similarity is used to compare matrices, the responses are assumed noise-free, and centering is applied to Representational Dissimilarity Matrices (RDMs) when using RSA.

- Result 1:  Linear CKA is equivalent to using squared Euclidean RDMs
- Result 2: Average squared canonical correlation coefficient, linear CKA (with a kernel involving whitening transformation), and squared Mahalanobis RDMs are equivalent
- Result 3: Non-linear CKA with Radial Bias Function (RBF) kernels is equivalent to squared Euclidean RDMs with elementwise non-linearity applied

Overall, the paper is very well written with clear mathematical proofs. With the advancements and increased interest in computing representational similarity in different fields, this paper shows that different fields are converging on equivalent metrics, and can benefit from cross-talk. Additionally, while these equivalences might be intuitive to researchers with a mathematical background, a clear demonstration was required to make it accessible to all researchers.

Suggestions that can improve the paper:
1. In the paper, the systems are assumed noise-free. I believe discussing if these results hold in noisy systems, what accommodations are required to extend these results to noisy systems, recommendations when comparing representational similarity in noisy systems, and the relevant literature would boost the usefulness and impact of the paper.
2. Typo: Line 227 "a RDM using eq(7). Many kernel functions".
3. Typo: Line 308 "are needed to achieve a good estimates"

---

### Official Review · Reviewer_MWgW · 2024-10-06
**Review of Submission 16**

**Rating:** 9
**Confidence:** 4

**Review:**

**Strengths:**
1. The writing is clear and lucid, and the relevant literature has been discussed adequately.
2. Documenting the similarity between different similarity metrics is particularly useful and relevant to this workshop, especially given the deluge of such metrics – it would be helpful to both practitioners and theoreticians to be aware of these equivalences, in order to guide their decision making when choosing similarity metrics for their work.

**Suggestions:**
I checked the mathematical results and wasn't able to spot anything amiss. Perhaps one "weakness" is that the paper treats neural responses as noise-free/non-stochastic, but this is not the case in neuroscience or in many works in theoretical/computational neuroscience. This is acknowledged in the paper but it might be interesting to have a brief discussion on metrics that take stochasticity into account, e.g., work from Duong et al. (2023). Nevertheless, this is not a negative -- several works consider deterministic neural networks or condition-averaged responses, and I reiterate that showing the equivalences between various metrics explicitly is useful.

Duong, L., Zhou, J., Nassar, J., Berman, J., Olieslagers, J., & Williams, A. H. Representational Dissimilarity Metric Spaces for Stochastic Neural Networks. In The Eleventh International Conference on Learning Representations.

I just have some minor suggestions on fixing typos/grammar issues in certain places (this list may not be exhaustive):
1. Line 28: "then use cosine similarity is used to compare" -> remove "is used"
2. Line 35: "Yanai's generalized generalized" -> remove duplicated "generalized"
3. Line 75: "Specify also a RDM" -> might be better to use "[an](https://blog.apastyle.org/apastyle/2012/04/using-a-or-an-with-acronyms-and-abbreviations.html) RDM"
4. Line 94: $\mathbf{C} = \mathbf{I} - \frac{1}{M} \mathbf{11}$ -> should be $\mathbf{11^\top}$, right?
5. Line 175: Should use `\citep` for reference 21, i.e., without explicit mention of "Kornblith et al." (unless you say "Kornblith et al. [21] advocated ..." in line 174).
6. Line 190: "_rsatoolbox_ package" -> "$\texttt{rsatoolbox}$ package" (to be consistent with Line 170)
7. Line 314: "practicioners" -> "practitioners"
8. Inconsistent usage of "I" and "we" throughout the paper: it might be better to just stick with "we" throughout

**Quality:** I think this is quality work with a good fit to this workshop. While it is not proposing something remarkably novel, this is a very useful, explicit documentation of some equivalences the UniReps community should be aware of (those with a strong mathematical background may already be familiar with these relationships, as the paper admits, but it is always beneficial to have explicit documentation of such knowledge). The math and overall presentation are great.

**Clarity:** As I stated earlier, I'm a fan of the writing of this paper and think it reads clearly.

**Originality:** While the work builds upon existing knowledge of equivalences between CCA and CKA from Kornblith et al. and Sejdinovic et al. ([21, 34] in the paper), I am not aware of any other work explicitly documenting the equivalences mention here.

**Significance:** This work is of great fit to UniReps and will be of interest/use to both the cognitive/computational neuroscience community and the machine learning community alike.

---

### Official Review · Reviewer_EMbc · 2024-10-07
**Instructive work bridging representation similarity measures from different communities**

**Rating:** 10
**Confidence:** 4

**Review:**

This contribution draws an equivalence between CKA and RSA. It outlines CKA, RSA, and CCA (canonical correlation analysis), which are representation similarity frameworks that take choices of, e.g., similarity or distance functions between neural responses. The authors show in a series of propositions that, under certain realizations (choices of the mentioned functions) of each framework, they produce identical scores on the same neural responses.

I found this work to be very clear and instructive. Given the frequent use of representation similarity frameworks now in the neuroscience and machine learning communities, this contribution can be a practical guide as to when similarity measures are equivalent. Given its timeliness and broad relevance to the workshop, I would strongly recommend the paper for acceptance.

---

### Decision · Program_Chairs · 2024-10-10

**Decision:**

Accept (Oral)

**Comment:**

In light of the positive reviewers' feedback and relevancy of the submission, we are pleased to accept this paper for presentation at UniReps 2024. We kindly ask the authors to incorporate the reviewers' suggestions and feedback in the final camera-ready version of the manuscript.